# Rates of Vaccination against COVID-19 in Psychiatric Outpatients

**DOI:** 10.3390/jpm14070748

**Published:** 2024-07-14

**Authors:** Mina Cvjetkovic Bosnjak, Dusan Kuljancic, Ana-Marija Vejnovic, Darko Hinic, Vladimir Knezevic, Dragana Ratkovic, Vanja Bosic, Vesna Vasic, Branislav Sakic, Darja Segan, Predrag Savic, Minja Abazovic, Masa Comic, Djendji Siladji, Dusica Simic-Panic, Olga Ivetic Poledica

**Affiliations:** 1Medical Faculty, University of Novi Sad, Hajduk Vejkova 3, 21000 Novi Sad, Serbia; ana-marija.vejnovic@mf.uns.ac.rs (A.-M.V.); vladimir.knezevic@mf.uns.ac.rs (V.K.); dragana.ratkovic@mf.uns.ac.rs (D.R.); vanjabosic11@gmail.com (V.B.); vzvasic@gmail.com (V.V.); branislav.sakic@gmail.com (B.S.); darja.segan@mf.uns.ac.rs (D.S.); predrag.savic@mf.uns.ac.rs (P.S.); minja.abazovic@mf.uns.ac.rs (M.A.); masa.comic@mf.uns.ac.rs (M.C.); djendjis@gmail.com (D.S.); dusica.simic-panic@mf.uns.ac.rs (D.S.-P.); olga.ivetic@mf.uns.ac.rs (O.I.P.); 2Clinics of Psichiatry, University Clinical Centre of Vojvodina, Hajduk Veljkova 4, 21000 Novi Sad, Serbia; 3PMF Kragujevac, University of Kragujevac, Radoja Domanovića 14, 34111 Kragujevac, Serbia; dhinic@kg.ac.rs; 4Clinics of Medical Rehabilitation, University Clinical Centre of Vojvodina, Hajduk Veljkova 4, 21000 Novi Sad, Serbia

**Keywords:** COVID-19, vaccination, mental disorders, anxiety, depression, somatic health

## Abstract

Background: The aim of this study was to compare the rates of vaccination against COVID-19 infection in psychiatric outpatients and the general population, as well as rates of infected patients. In addition, the level and type of anxiety due to the pandemic were observed in patients with psychotic, anxiety, and depressive disorders. Materials and Methods: In the present study, 171 patients with pre-existing mental disorders completed the questionnaire about the doses and types of vaccination against COVID-19. During 2021–2023, patients with different mental disorders, aged from 18 to 80, were included. All patients filled in a self-reported questionnaire including general information (age, sex, marriage, education, working status, comorbid conditions) as well as questions about mental health, receiving vaccination, and the course of COVID-19 infection if it was present. All patients gave informed consent for the interview. Results: Patients with pre-existing mental disorders were more likely to be vaccinated against COVID-19 compared with the general population. The Sinopharm vaccine was most frequently applied. In the observed patients, 46.8% were infected, but just 7% had a medium or serious form of infection and were not vaccinated. Conclusions: In our study, the percentage of vaccinated psychiatric patients was greater than that in the general population, except in psychotic patients, who were mostly limited by fear. Such results can be explained by the high percentage of somatic comorbidities in this population and perhaps insufficient information about the positive effects of vaccination.

## 1. Introduction

In recent years, the world has had to face enormous, life-threatening situations such as pandemics, wars, earthquakes, and storms. The mental health of the general population of all ages and genders was often negatively impacted when facing such terrifying events. The COVID-19 pandemic represents one of greatest stress events to have affected almost every citizen on Earth, and even though it has passed, it still instills fear in people who were or were not infected and who lost loved ones, as well raising awareness in most people that such disasters could occur again. Considerable current evidence suggests that the risk of various subtypes of coronavirus persists, and healthcare workers worldwide need to be ready to properly react in a timely manner should another epidemic/pandemic take place.

Becoming infected with the COVID-19 virus itself endangers health and affects personal security, social integrity, and general functioning. This, along with the restrictive epidemiological measures in the form of restrictions on movement, social isolation and distancing, and the prevention of physical contact, although effective in reducing the rate of transmission and infection, can together cause a state of heightened personal and collective psychological tension, accompanied by frequent anxiety and a sense of fear, especially of death, and a sense of finality and endangerment of personal health [1,2]. The most stressful aspects of public health crises such as pandemics are their unpredictability, as well as the uncertainty regarding disease control and the assessment of the degree and severity of health risks [1].

Uncertainty, constant challenges, and stress in crisis situations such as epidemics and pandemics of infectious diseases can have a negative effect on mental disorders by inducing them and complicating their course and outcome [3]. The pandemic caused by the SARS-CoV-2 virus itself, as well as all the epidemiological measures introduced to contain it, represents a psychological burden for the population, disrupting personal, family, and social functioning in the individual. This is especially true in vulnerable social groups, such as psychiatric patients, who are often on the margins of society under normal circumstances [3]. According to experts’ estimates, effects will reach their peak in the near future and the effects of the current pandemic are likely to persist for a long time [4]. In the research by Brooks et al., it is pointed out that periods of self-isolation, restriction of social contact, and quarantine, even for less than 1 day, can have long-term consequences with the presence of mental disorders up to 3 years later [5]. Considering these factors, extremely stressful situations such as the COVID-19 pandemic can trigger new mental disorders in the ways described above, as well as exacerbate existing pathopsychological factors on biological and psychological levels.

Global studies in the early stage of the pandemic found that patients with mental health disorders were at higher risk of COVID-19 infection and had increased rates of somatic and neuropsychiatric complications and mortality [6,7]. COVID-19 has been associated with adverse effects on the brain, including dizziness, headache, impaired consciousness, and cerebrovascular complications [8], with studies suggesting that patients with mental illness suffer additional symptoms such as delirium, altered consciousness, and encephalitis [9,10]. Emerging evidence suggests that individuals with serious mental illness have an increased risk of infection due to impaired cognition and institutional barriers to health care [9,10].

During the pandemic, considerable evidence emerged suggesting that psychiatric patients were subjected to a more severe course of COVID-19 infection compared with the general population [11,12,13,14]. Such results motivated this study. It is known that the pandemic itself, the fear and stress that accompanied it, and the threat to physical and psychological integrity were factors contributing to the disruption to personal well-being without any difference in relation to the existence of mental illness [15]. Furthermore, as psychiatric patients are often on the margins of society [15], and vaccination is a method of primary prevention, we attempted to explore the attitudes and outcomes in psychiatric patients in relation to COVID-19 vaccination, and issues relating to its availability.

In Serbia, the COVID-19 pandemic caused by SARS-CoV-2 was announced officially on 11 March 2020. Pandemics of infectious diseases usually affect both somatic and mental health in the general population as well as in psychiatric patients. In the population of psychiatric patients, there have been few investigations about attitudes to vaccination, the rate of vaccination, and the course of illness in people with pre-existing mental disorders [11,12,13].

Some authors report lower rates of vaccination among patients with psychiatric disorders, especially among patients with schizophrenia [12,13,14,16]. There are also some predictions that patients with psychiatric disorders, if infected, will demonstrate more serious types of infection and usually have a complicated course of infection [11,16,17,18]. In an overview of the literature and in data available online, there is also evidence that patients with some psychiatric disorders, like non-affective psychosis, had lower rates of vaccination, as they had fewer social contacts and were unsure about the positive effects of vaccination. This was due to personal changes as well as paranoid interpretations of vaccination and a lack of motivation [11,12,19,20,21,22].

The psychological aspects behind the lower vaccination rate in psychiatric patients mostly depend on the very psychopathological basis of the underlying mental illness. Paranoid patients are more susceptible to the influence of the media and misinformation that spreads through it about the possible harmfulness of vaccination, such as confirming delusional thoughts about conspiracies against the population by governments and multinational companies to reduce and control the number of people on Earth. This can be the reason for categorically refusing vaccination in such patients, while in anxious patients, all the stressful situations of the pandemic represent a trigger for anxiety of pathological proportions, which is paralyzing, and they thus remain unvaccinated. Depressed patients are by nature disinterested and suicidal, have given up on themselves and their health, and are unmotivated, and they surrender to fate and thus remain unvaccinated. In cases of organically determined mental disorders, healthcare workers often failed to address patients on waiting lists in many hospital wards due to the overload of the health system during the pandemic, and therefore, these patients also remained unvaccinated [18,19].

These are possible reasons why, according to the literature, psychiatric patients remain unvaccinated. Basically, as demonstrated in this paper, the stress and situations caused by the pandemic exacerbated the basic symptoms of psychiatric diseases.

Some recent investigations have shown that patients who suffered from different psychiatric illnesses had lower rates of vaccination than people in the general population [12,14,17,23], and that such patients were at greater risk of being infected by SARS-CoV-2 or had more severe clinical signs compared with the general population [20,23,24]. In addition, there is evidence that mortality during the pandemic due to COVID infection was higher in patients suffering from psychiatric illness (most frequently accompanied with lower socioeconomic status and various comorbid somatic problems) [25].

The aim of this investigation was to compare the rates of vaccination against COVID-19 infection in psychiatric outpatients compared with the general population, as well as the rate of infected patients. Furthermore, the level and type of anxiety due to the actual pandemic were observed in patients with psychotic, anxiety, and depressive disorders and their links to the vaccination rate were investigated.

Finally, our hypothesis is that a significant percentage of outpatient psychiatric patients are vaccinated compared with the general population in Serbia. In addition, the assumption was that the level of anxiety would be higher in our sample of psychiatric patients compared with the general population, but that this would not negatively affect the vaccination rate in the studied population.

## 2. Materials and Methods

This was a retrospective cross-sectional study in which 171 outpatients with pre-existing mental disorders were observed. All patients were selected from one psychiatric institution, Clinic of psychiatry, University Clinical Centre of Vojvodina, Novi Sad from 2021 to 2023. Patients aged 18–70 fulfilled the criteria for either affective disorders or non-affective psychotic disorders. All patients who met the criteria for inclusion in this study were included, and no statistical methods were applied to form the sample.

The first part of the questionnaire collected data about diagnosis, gender, age, marital status, education, working status, comorbid conditions, and pre-existing psychiatric disorders. The second part of the questionnaire collected data about vaccination motivation, type of vaccine, evidence of positive PCR tests for SARS-CoV-2, and the type and intensity of anxiety symptoms connected to the pandemic (FCV-19S Scale). Patients filled in the questionnaires on their own on admission and during regular check-ups. The Fear of COVID-19 Scale (FCV-19S Scale) is a seven-item unidimensional scale with robust psychometric properties. More specifically, its reliability values such as internal consistency (α = 0.82) and test–retest reliability (ICC = 0.72) were acceptable. The total scores on the FCV-19S were comparable across both genders and all ages, which suggests that it was a good psychometric instrument for use in assessing and allaying fears of COVID-19 among individuals. Higher scores on this scale suggested a higher degree of fear of the COVID-19 pandemic and vice versa. The scale is available in the public domain [26]. Data about the type and doses of the administrated vaccine and the course of disease in infected patients as well as the type and intensity of anxiety symptoms were noted. All patients gave informed consent for the interview. The Ethical Board of University Clinical Centre of Vojvodina approved the conduct of this study in the document under registration No. 7/21.

SPSS for Windows 20 was used for data processing, running on Microsoft Windows. The results were tabulated and are presented graphically in this paper. Descriptive statistics (frequencies and percentages for categorical data) are presented. The χ² test was applied for categorical data to determine the connection between the psychiatric diagnosis (psychotic, depressed, anxious) and whether the patient was treated for coronavirus, whether they contracted coronavirus, whether they were vaccinated, how many doses they received, and whether COVID-19 was a trigger for their worsening. To determine the frequency of vaccine doses and intensity of anxiety among the three groups of patients, cross-tabulation was applied. Regarding the variables comorbidity and COVID-19 anxiety, as patients provided more answers, the analysis of percentages determined which type of comorbidity occurred most often in the three groups of patients and which factors contributed most to COVID-19 anxiety.

## 3. Results

In the present study, 89.1% of patients were female and 19.9% were male. In the whole sample, 16.9% of patients had psychotic disorders, 46.2% had a depressive disorder, and 36.9% had anxiety disorders (Table 1). Almost half of the patients were middle-aged (52.4%). Most patients were employed (46.7%), 12% were unemployed, and 41.3% were retired (Table 1). Most patients lived with family members (71.9%). More than 70% had high school education.

More than 70% of patients were psychiatric patients before the pandemic (Table 2) and 25.1% were not vaccinated—mostly patients with psychotic disorders. A high percentage of patients were vaccinated (74.7%). In the group of vaccinated patients, most received the Sinopharm vaccine (61.7%), 17.1% received the Pfizer vaccine, and 11.2% received the Sputnik vaccine. More than half of vaccinated patients were completely immunized with all three doses. In this sample, 46.7% of patients had had a positive PCR test for COVID-19; infection was present mostly in unvaccinated patients (Table 3).

A third of the patients with psychotic and anxiety disorders and a quarter of the patients with depression did not receive any dose of the vaccine. A single vaccine dose was received by 7% of patients with psychotic disorder and less than 5% of patients with depression and anxiety disorders. Two doses were received by 17% of patients with psychotic and depression disorders and 11% of patients with anxiety disorder. All three doses were received by 55% of patients with depression and anxiety disorders and 40% of patients with psychotic disorder (Table 3 and Table 4).

In this sample, 87.1% of patients had one or more chronic somatic problems (Table 5), mostly hyperlipoproteinemia (64.4%), cardiovascular problems (47%), obesity (43.6%), hypothyreosis (17.4%), and diabetes mellitus (22.1%). All comorbidities were medically treated.

There is statistically significant evidence that during the pandemic, more patients had primal manifestations of anxiety disorders compared with patients with depression or psychotic disorders (χ² = 12.963, *p* = 0.02). Most patients with depressive and psychotic disorders had already been receiving treatment before the pandemic. Cramer’s V = 0.276 indicated a moderate correlation between variables using the −χ² test of association between pre-pandemic treatment and psychiatric diagnosis (Table 6).

There were no differences between patients with anxiety, depression, and psychosis in regard to frequency of infection. About 50% of depressive patients were infected, similar to the group with psychotic and anxiety disorders, and 40% of total patients were COVID-19-positive—χ² (*n* = 171) = 1.289, *p* = 0.525 (Table 7).

A questionnaire on the intensity of anxiety symptoms was created for this investigation (Table 8). The level of anxiety (fear, anxiety, trembling, palpitation, insomnia) was measured from 1 to 10 in accordance with patients’ statements during the pandemic: mild (1), medium (2–4), moderate (5–7), severe (8–9), and very severe (10).

In the investigated patients, the level of anxiety was high. In most patients (92.5%), anxiety symptoms were, according to subjective feelings, severe or very severe (Table 8). About 90% of all patients felt fear for themselves, their parents, or close members of their families, and 40% were worried about unemployment and about economic consequences of the pandemic. Severe and very severe intensities of anxiety were recorded in nearly half of the psychotic and anxious patients (Table 8).

## 4. Discussion

The literature data show a lot of evidence that patients with psychiatric disorders are a vulnerable category and present with more serious forms of coronavirus disease if they become infected [27,28]. The reason for this greater risk lies in the symptoms of disorders, such as a loss of energy and will; patients very often do not take care of their somatic health, avoid primary care physicians, do not have health insurance, etc. [16,22,29]. Stigmatization, emotional symptoms, cognitive deficits, and impaired communication usually have negative effects on receiving adequate and continuous primary health care in these patients [12,20,29,30].

Therefore, the aim of this study was to examine the rate of vaccination against COVID-19 in psychiatric patients in Serbia in relation to their diagnosis and in relation to somatic comorbidities, as well as to examine the impact of the fear of COVID-19 infection on the decision to be vaccinated. Our intention was to examine the attitudes of psychiatric patients to vaccination and to take important measures to prevent mass epidemics of infectious diseases, considering the amount of misinformation and quasi-scientific attitudes that are abundant in the public media. In this way, we wish to highlight a psychiatrist’s holistic concern for the overall health of their patients and their very important role in that field.

The results of this study show that patients with depression and anxiety disorders were more likely to be vaccinated compared with the general population and patients with non-affective psychotic disorders. In addition, patients with somatic comorbidity had higher levels of vaccination compared with the group without comorbidities. These results suggest that patients were well informed about the positive effects of vaccination, especially those with somatic comorbidities. The fact that patients with psychotic disorders were less willing to be vaccinated can be interpreted according to the types and characteristics of mental disorders [30,31,32,33]. The fear shown by people with mental disorders in relation to the COVID-19 pandemic is more pathological in nature and etiology compared with people without mental disorders [15,34,35]. The feeling of fear can be classified as sthenic or asthenic depending on etiology, personality type, and the type of psychiatric disorder present. According to the results of studies worldwide, as well as studies in our country, the level of fear caused by the COVID-19 pandemic in both the general population and psychiatric patients is very high [15,36]. High levels of fear in patients with psychotic disorders of non-affective etiology very often have paralyzing characteristics, reaching psychotic levels and inducing psychotic symptoms of underlying mental health disorders [34]. In such a state of psychotic fear, it is easy to understand why psychiatric patients with psychotic disorders do not accept vaccination. In a state of increased social tension, insecurity, and threats to mental and physical integrity, while on the other hand being surrounded by misinformation and pseudoscientific attitudes about vaccination, psychotic patients very easily develop an almost insurmountable resistance to vaccination, while also being under the influence of delusional thoughts and perceptual disturbances that arise on the basis of illness in a paranoid environment and situation [18].

In our opinion, the psychotic quality of fear in psychotic patients, caused primarily by misinformation, is the main reason why they were reluctant to accept vaccination. On the other hand, in anxious and depressed patients, the appearance of fear, especially for their own physical health and integrity, and the possibility of infecting their closest family members had a sthenic effect, and this group of patients, trusting primarily their psychiatrists, accepted vaccination against COVID-19 ahead of all other social groups. In our opinion, the factors affecting the decision about vaccination include proper health education and provision of scientifically informed information about this preventive method, especially for patients who are mentally ill, which is not an easy task at all.

Although there have been few studies reporting higher rates of vaccination in mentally ill patients, the overview of the literature shows considerable evidence that the vaccination rate among these patients is significantly lower compared with the general population [12,13,15,18,25,31]. It can be considered that the psychiatric patients involved in this study, due to the fact that they were in the process of treatment and psychiatric monitoring, had easier access to vaccination against COVID-19. They had access to doctors to influence them in terms of health education and perhaps, because of this, they were more likely to be vaccinated compared with the general population. However, this argument cannot fully stand. It may be that this only applies to permanently institutionalized psychiatric patients, as a separate subcategory. In Serbia and also throughout the world, there is stigma in relation to psychiatric patients. They are marginalized in accessing and using health care in terms of somatic treatment. Thus, the argument that psychiatric patients in the outpatient environment that we followed were more frequently vaccinated just because they were in the system does not stand and cannot be applied in our case. Psychiatric patients, unfortunately, according to practical experience, have never been a priority group in the care of somatic health in healthcare systems worldwide [37]. Therefore, we believe that the basis of such results that our study reached underlines the deeper psychological mechanisms associated with the stress and fear that accompanied the COVID-19 pandemic itself, which were especially pronounced in psychiatric patients

As a consequence, it is very important and must be of priority interest for psychiatrists to help such patients in overcoming barriers and in improving overall health. Timely and complete vaccination against COVID-19 is necessary for reducing the risk of serious infections and the high level of mortality [38].

The vaccination rate in our patients was high, even higher than in the study from Greece, where the rate was slightly below 70% [38]. The difference is that the rate of vaccination is different in Serbia in psychiatric patients compared with the general population, while in Greece and in other countries in our immediate or wider environment, such a finding has not been observed. In Serbia, according to the official results [39], slightly more than 50% of the population were vaccinated, while in our sample of psychiatric patients, the vaccination rate was as much as 25% higher and was around 75% in total, which is a very significant value and a statistically significant difference compared with the general population.

This discrepancy in the results can be attributed, on the one hand, to a large amount of misinformation in the general public, as well as to a significant anti-vaxxer lobby that managed to reach the masses, but on the other hand, to the role of psychiatrists in the Clinical Center of Vojvodina who, through their personal commitment to health promotion, succeeded in presenting to the vast majority of their patients all the benefits of vaccination against COVID-19 and in disproving all the taboos related to the same topic in order to encourage them to respond to the call for vaccination in ever greater numbers, which this study shows.

The study by Greek authors showed no difference between the rates of vaccination in relation to the diagnosis of mental disorders [38]. In all groups of disorders, the percentage of the vaccinated was equal. However, in our sample, patients with psychotic disorders were significantly less vaccinated than those suffering from anxiety and affective disorders. The vaccination rate against COVID-19 in these patients in our study was about 66%, which was lower than the overall rate of vaccinated psychiatric patients, but still higher than in the general population.

Such findings are partly the result of the very nature of the psychotic process, where patients are markedly suspicious, cautious, distrustful, paranoid, socially withdrawn, passive, and unable to take care of their own health [28], but on the other hand, there was misinformation that we witnessed through all information systems, which significantly influenced the increase in fear and insecurity in all the population, especially in psychiatric patients, in whom fear often had pathological characteristics, to which psychotic patients are especially prone [15].

Nevertheless, future research is needed to address the limitations of this study. The first limitation was due to the fact that this study utilized a self-filled questionnaire-based survey, and all variables were assessed through self-reported measures, which are susceptible to various biases, such as social desirability bias, as well as inaccuracies due to memory errors or subjective interpretations of the questions. The current survey could also have limited the sample’s representativeness to patients from only one mental health institution. This could have led to the sample not being fully representative of the broader population, thus limiting the generalizability of the current findings. In addition, while self-reported surveys are convenient and cost-effective, they often lack the depth and nuance that can be obtained through other methods, such as in-depth interviews or observational studies. These alternative methods can provide richer, more detailed data that can more effectively capture the complexities of the participants’ experiences. To address these limitations, future research should consider incorporating a mix of methodologies, such as combining self-reported surveys with objective measures and qualitative methods. Additionally, efforts should be made to ensure a more diverse and representative sample, potentially through targeted recruitment and widening the sample sources to other institutions in Serbia.

We believe that special attention should be paid to the feeling of fear that the COVID-19 pandemic itself causes in all of us, especially in the population of psychiatric patients, as we know that the feeling of fear itself may affect our activities, but it can also be something that holds us back and hinders us. In addition, the fear itself arises from ignorance and insufficient and poor information, which was present during the COVID-19 pandemic in Serbia and worldwide. Psychotic patients were particularly susceptible to such misinformation, which in their case found fertile ground for the development of paranoid ideas and resistance to vaccination. However, there was a noticeable increase in the incidence of anxiety disorders during the pandemic. In our study, the number of primary manifestations of anxiety disorders during the pandemic was significant. In such patients, a lower vaccination rate was also observed compared with patients with affective disorders. Here, again, we emphasize the high levels of anxiety recorded in anxious patients during the pandemic, both in our country and throughout the world, as contributing factors to the appearance of fear and resistance to vaccination. Our conclusions are in line with several studies from around the world [40,41,42,43,44,45,46].

## 5. Conclusions

This investigation showed that the rate of vaccination against COVID-19 was higher in patients with depressive and anxiety disorders who were willing to be vaccinated than in the general population. Psychotic patients are the least motivated to be vaccinated and are therefore most exposed to the disease, all because of the high level of fear that induces delusional thoughts about vaccination in these patients. The level of fear in our examined patients was higher than that in the general population and, as such, had a negative effect on the level of vaccination, primarily in psychotic patients, often inducing a bout of psychosis that caused vaccination refusal. Only half of the included patients were infected, without complications of infection. There are multiple possible reasons for the more serious course of infection in psychiatric non-vaccinated patients; such patients often neglect somatic symptoms and have less contact with primary care, and on the other hand, there is a possibility that physicians tend to focus on mental problems in psychiatric patients and, due to stigmatization, neglect somatic problems.

## Figures and Tables

**Table 1 jpm-14-00748-t001:** Sociodemographic data.

Variable	No	(%)
Gender		
Female	137	80.1
Male	34	19.9
Type of mental disorder		
Psychotic	29	16.9
Depressive	79	46.2
Anxiety	63	36.9
Age in years		
20–30	7	4.1
31–40	27	15.9
41–50	89	52.4
51–60	41	24.1
>65	6	3.5
Level of education		
Elementary school	13	7.6
High school	122	71.3
College	36	21.1
Employed		
Yes	78	46.7
No	20	12.0
Retired	69	41.3
Living with family members		
Yes	120	71.9
No	47	28.1

**Table 2 jpm-14-00748-t002:** Questions about COVID-19 infection and vaccination.

Item	No	%
Being a psychiatric patient pre-pandemic		
yes	123	71.9
no	47	28.1
COVID-19 positive test		
yes	77	46.7
no	94	53.3
Vaccination against COVID-19		
yes	128	74.7
no	43	25.3
Type of vaccine		
Sinopharm	79	61.7
Pfizer	29	23.4
Sputnik	19	14.8
No. of doses		
one	6	4.7
two	30	23.4
three	92	71.8
Pandemia COVID-19 as a trigger for mental disorder episode		
yes	100	58.5
no	71	41.5

**Table 3 jpm-14-00748-t003:** χ² test correlation between vaccination and type of mental disorder.

Type of Mental Disorder	N	Vaccination, Type				χ²	*p*
		Not Vaccinated	Sinopharm	Pfizer	Sputnik		
Patients with psychotic dis.	29	7	13	5	4	2.329	0.887
Patients with depressive dis.	79	19	39	11	10		
Patients with anxiety dis.	63	17	29	13	5		
Total	171	43	79	29	19		

Legend: N—number of patients; χ²—statistics; *p*—statistical significance.

**Table 4 jpm-14-00748-t004:** Cross-tabulation doses of vaccination and type of mental disorders.

Type of Mental Disorder	N	Doses			
		0	1	2	3
Patients with psychotic dis.	29	10	2	5	12
Patients with depressive dis.	79	20	2	14	43
Patients with anxiety dis.	63	19	2	7	34
Total	171	49	6	26	89

Legend: N—number of patients; χ²—statistics; *p*—statistical significance.

**Table 5 jpm-14-00748-t005:** Types of somatic comorbidities in the sample.

Item	Patients (*n* = 171)	%
Patients with somatic comorbidities		
Yes	149	87.1
No	22	12.9
More than one somatic comorbidity		
Yes	120	80.5
Hyperlipoproteinemia and hypercholesterolemia	96	64.4
Cardiovascular	70	47
Overweight	65	43.6
Diabetes mellitus	33	22.1
Hypothyreosis	26	17.4

**Table 6 jpm-14-00748-t006:** The prevalence of newly discovered disorders and exacerbations present pre-COVID-19 pandemic.

Diagnosis	Patients (*n* = 171)	Being a Psychiatric Patient Pre-Pandemic		χ²	*p*
		Yes	No		
Psychotic disorders	29	26	3	12.963	0.002
Depressive disorders	79	62	17		
Anxiety disorders	63	36	27		
Total	171	124	47		

Legend: N—number of patients; χ²—statistics; *p*—statistical significance.

**Table 7 jpm-14-00748-t007:** χ^2^ test correlation between positive PCR test and psychiatric diagnoses.

Diagnosis	PCR+ for COVID-19		χ²	*p*
	Yes	No		
Psychotic disorders (*n* = 29)	12	17	1.289	0.525
Depressive disorder (*n* = 79)	39	41		
Anxiety disorders (*n* = 63)	27	35		
Total	78	93		

Legend: N—number of patients; χ²—statistics; *p*—statistical significance.

**Table 8 jpm-14-00748-t008:** Intensity of anxiety symptoms associated with COVID-19.

Type of Mental Disorder	Intensity of Anxiety Symptoms				Prevalence of Severe and Very Severe Intensity of Anxiety %
	Mild	Moderate	Severe	Very Severe	
Patients with psychotic disorders (*n* = 29)	1	15	10	3	44.8
Patients with depressive disorders (*n* = 80)	2	47	27	4	38.75
Patients with anxiety disorders (*n* = 62)	0	33	23	6	46.77
Intensity of anxiety symptoms	Patients				
	N = 171		%		
Mild	3		1.8		
Moderate	95		55.5		
Severe	60		35.0		
Very severe	13		7.6		

## Data Availability

The data that support the findings of this study are available from the corresponding authors (M.C.B. and D.K.) upon reasonable request.

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
