# Peer review of "Rates of Vaccination against COVID-19 in Psychiatric Outpatients"

_jpm, 2024, doi:10.3390/jpm14070748_

Round 1

Reviewer 1 Report

Comments and Suggestions for Authors

I have carefully reviewed the manuscript, titled “Rate of vaccination against COVID-19 in psychiatric outpatients”. The study was to compare rates of vaccination against Covid-19 infection in psychiatric outpatients and general population as well as rate of infected patients.

Although the study presents some arguments, a number of points need to be addressed.

Introduction:

1) The Introduction section should be more detailed and precise, with more information on factors affecting mental disorders.

2) Can you present psychological and medical mechanisms responsible for rates of vaccination in psychiatric outpatients? What are its underlying mechanisms (p. 2).

3) It would be beneficial to provide more information on the association of COVID-19 and mental health.

4) Hypotheses need to be formulated to specify the aim of the study.

Method:

5) Was the sample determined by power analysis?

6) There should be psychometric values for the questionnaires used in your study.

7) How did you handle missing values in your data? (If any exist)

Results:

8) The results need to be more broadly presented (pp. 5-8).

Discussion:

9) What are the underlying mechanisms responsible for this result: “patients with depression and anxiety disorders were more likely to be vaccinated, compared to general population and patients with non-affective psychotic disorders”? (p. 9).

10) The results described on p. 9 should be more thoroughly discussed in the context of mental health.

 11) This statement should be elaborated on: “The fact that patients with psychotic disorders are less willing to get vaccinated can be interpreted with the type and characteristics of mental disorders” (p. 9). How would you explain it?

12) What are the limitations of the study? (p. 10).

Comments on the Quality of English Language

I have carefully reviewed the manuscript, titled “Rate of vaccination against COVID-19 in psychiatric outpatients”. The study was to compare rates of vaccination against Covid-19 infection in psychiatric outpatients and general population as well as rate of infected patients.

Although the study presents some arguments, a number of points need to be addressed.

Introduction:

1) The Introduction section should be more detailed and precise, with more information on factors affecting mental disorders.

2) Can you present psychological and medical mechanisms responsible for rates of vaccination in psychiatric outpatients? What are its underlying mechanisms (p. 2).

3) It would be beneficial to provide more information on the association of COVID-19 and mental health.

4) Hypotheses need to be formulated to specify the aim of the study.

Method:

5) Was the sample determined by power analysis?

6) There should be psychometric values for the questionnaires used in your study.

7) How did you handle missing values in your data? (If any exist)

Results:

8) The results need to be more broadly presented (pp. 5-8).

Discussion:

9) What are the underlying mechanisms responsible for this result: “patients with depression and anxiety disorders were more likely to be vaccinated, compared to general population and patients with non-affective psychotic disorders”? (p. 9).

10) The results described on p. 9 should be more thoroughly discussed in the context of mental health.

 11) This statement should be elaborated on: “The fact that patients with psychotic disorders are less willing to get vaccinated can be interpreted with the type and characteristics of mental disorders” (p. 9). How would you explain it?

12) What are the limitations of the study? (p. 10).

Author Response

Dear Reviewer,

I would like to first of all thank you on behalf of my research team for the time you took to review our scientific work. We are really grateful for all the constructive comments that together make our work far more qualitative and more receptive and reasonable to the scientific and wider audience, which is of particular importance considering the seriousness and topicality of the topic of the work.

We have taken into account all your valued remarks and I hope that we have responded to them in an adequate way and thus made it worthwhile to improve the quality of the work and make it acceptable for publication.

In the following text, I will present detailed explanations of your requested corrections.

  1. „The Introduction section should be more detailed and precise, with more information on factors affecting mental disorders.“-The introduction section has been supplemented, reworked, changed so that now I hope it is richer, clearer, more precise and explains which factors related to the COVID-19 pandemic affect mental health, both in general and in those with mental illness before. Also, the assumed psychological background regarding the motivation and willingness to vaccinate in relation to psychological factors is presented.
  2. „Can you present psychological and medical mechanisms responsible for rates of vaccination in psychiatric outpatients? What are its underlying mechanisms (p. 2).“-We sought to describe the hypothesized psychological and medical mechanisms influencing the decision of psychiatric outpatients to get vaccinated against COVID-19.
  3. „It would be beneficial to provide more information on the association of COVID-19 and mental health.“-Furthermore, we provided more data on the impact of the COVID-19 pandemic on mental health both in the general population and in psychiatric patients.
  4. „Hypotheses need to be formulated to specify the aim of the study.“-Hypotheses are formulated according to the aim of the study and they are presented at the end of the Introduction section.
  5. „Was the sample determined by power analysis?“-The sample in our study was not determined by any power analysis. All patients who met the criteria for inclusion in the study and who visited our psychiatric hospital in a given period of time as written in the research methodology were included in the research.
  6. „There should be psychometric values for the questionnaires used in your study.“-Psychometric values for the questionnaires that are validated are included in the mainn text now.
  7. „How did you handle missing values in your data? (If any exist)“-There are no missing values in our data.
  8. „The results need to be more broadly presented (pp. 5-8).“- I hope that we managed to present our results in a better and more comprehensive way. Now it is clearer to us what we got and we give a clearer confirmation of our conclusions. I have to inform you that the second reviewer had a request to summarize and rationalize the presentation of the results, and you requested to expand it. On this point, your two opinions are in conflict. We, as authors, have decided to expand the presentation of the results, but to try to standardize the format of the tables and to make a certain selection of data in the results. I hope our solution will be satisfactory.
  9. „What are the underlying mechanisms responsible for this result: “patients with depression and anxiety disorders were more likely to be vaccinated, compared to general population and patients with non-affective psychotic disorders”? (p. 9).“- The underlying psychological mechanisms responsibile for our claim are listed in the discusion section and are supported with literature data.
  10. „The results described on p. 9 should be more thoroughly discussed in the context of mental health.“- Whole Discusion section is broaden and presents more detailed opinion on our results in the context of mental health and problems affecting it in accordance with the COVID-19 pandemics.
  11. „This statement should be elaborated on: “The fact that patients with psychotic disorders are less willing to get vaccinated can be interpreted with the type and characteristics of mental disorders” (p. 9). How would you explain it?“- A detailed explanation is provided in the main text.
  12. „What are the limitations of the study? (p. 10).“- A whole new paragraph about the limitations of the study is added in the end of Discusion section. We are thankful on this remark.

I hope we have met your requirements and that our solutions will be satisfactory enough to make this paper suite for publication.

Looking forward to hearing good news from you soon.

Sincerely Yours,

Authors

Reviewer 2 Report

Comments and Suggestions for Authors

Dear authors,

Thank you for your submission. Your study, considering the topic, seems relevant to the journal. I will leave some comments for improvement:

Line 25: You do not need two decimal places when writing percentages. Include only significant digits

Lines 89-90: Please mention the registry number of the ethical approval document.

All tables: Format them to improve the structure. It is thin and has plenty of space. Furthermore, standardize the fonts using MDPI's recommendation. Also, there are too many tables with few comments between them. Find a way to better summarize the data and consider using plots in some cases.

Lines 184 to 189: I think you should discuss your findings (lines 190-196) before discussing other authors' findings.

Lines 190-196: I think your argument has room for more critical reflection. For instance, perhaps the patients were more likely to be vaccinated because they were a priority group or because, being patients, they were already under care and it was easy for doctors to influence them. I recommend a more critical analysis of the situation. Furthermore, are patients with the disorders under study always able to consent to vaccination?

Other than the comment above, you did good work and most of your discussion is highly valuable for the understanding of the topic. For a major public health issue like COVID-19, attention to vulnerable groups is vital.

Yours sincerely

Comments on the Quality of English Language

Dear authors,

Regarding the writing, I leave the following comments:

Line 14: Change "Covid-19" to "COVID-19". Review this in the entire document. COVID-19 should be in capital letters, just like AIDS or PTSD.

Line 18: by "fulfil questionare" did you mean "filled a questionnaire"? Review also the entire document to fix this.

Line 22: change "approvement" to "approval".

Perhaps you need to do a thorough grammar and spelling review of the entire document, if possible with an English native or advanced-speaking person.

Yours sincerely

Author Response

Dear Reviewer,

I would like to first of all thank you on behalf of my research team for the time you took to review our scientific work. We are really grateful for all the constructive comments that together make our work far more qualitative and more receptive and reasonable to the scientific and wider audience, which is of particular importance considering the seriousness and topicality of the topic of the work.

We are grateful for your praise for our work and effort, and we are grateful that you recognize the importance of the topic we deal with in the comprehensive health care of psychiatric patients who, as we emphasize, are a marginalized social group even under normal circumstances, let alone during such a major public health crisis as a pandemic COVID-19.

We have taken into account all your valued remarks and I hope that we have responded to them in an adequate way and thus made it worthwhile to improve the quality of the work and make it acceptable for publication.

In the following text, I will present detailed explanations of your requested corrections.

  1. „Line 25: You do not need two decimal places when writing percentages. Include only significant digits“- It is corrected.
  2. „Lines 89-90: Please mention the registry number of the ethical approval document.“- The registry No of ethical approval document is mentioned in the ain text.
  3. „All tables: Format them to improve the structure. It is thin and has plenty of space. Furthermore, standardize the fonts using MDPI's recommendation. Also, there are too many tables with few comments between them. Find a way to better summarize the data and consider using plots in some cases.“- Dear reviewer, we had the most difficulty with this remark of yours, since you and the first reviewer are contradictory when it comes to presenting the results. The first reviewer requested, for example, that the result display be expanded and supplemented with n+other tables. We standardized the tables, edited them as much as we could, but due to the request of the first reviewer, we still did not decide to reduce the number of tables. A more detailed presentation with a discussion of the results is given in the Discussion section. That's why we decided on a tabular presentation of the results. I hope you are satisfied with our solution.
  4. „Lines 184 to 189: I think you should discuss your findings (lines 190-196) before discussing other authors' findings.“- We are grateful for this remark. The order of presentation is changed as you requested.
  5. „Lines 190-196: I think your argument has room for more critical reflection. For instance, perhaps the patients were more likely to be vaccinated because they were a priority group or because, being patients, they were already under care and it was easy for doctors to influence them. I recommend a more critical analysis of the situation. Furthermore, are patients with the disorders under study always able to consent to vaccination?“- An interesting observation. However, in the text, we tried to refute it with arguments and show that the basis of the discrepancy regarding the vaccination rate against COVID-19 in the group of psychiatric patients and the general population is not a case and a random set of circumstances, as you thought, but that it is a matter of serious psychological mechanisms in the background that were triggered by the stressful circumstances that followed the pandemic.
  6. As far as the comments about the quality of English language are concerned, all you have poited we have managed accordingly. The Editor suggested a proofreading of the entire text by the proofreading service within MDPI, which we will accept. Therefore, the quality of the English language will be improved.

I hope we have met your requirements and that our solutions will be satisfactory enough to make this paper suite for publication.

Looking forward to hearing good news from you soon.

Sincerely Yours,

Authors

Round 2

Reviewer 1 Report

Comments and Suggestions for Authors

The authors have adhered to my suggestions and significantly improved their paper, so I recommend it for publication.